# Allosteric Modulation of Muscarinic Receptors by Cholesterol, Neurosteroids and Neuroactive Steroids

**DOI:** 10.3390/ijms232113075

**Published:** 2022-10-28

**Authors:** Ewa Szczurowska, Eszter Szánti-Pintér, Alena Randáková, Jan Jakubík, Eva Kudova

**Affiliations:** 1Institute of Organic Chemistry and Biochemistry, Czech Academy of Sciences, Flemingovo Namesti 2, Prague 6, 166 10 Prague, Czech Republic; 2Institute of Physiology, Czech Academy of Sciences, Videnska 1083, 142 20 Prague, Czech Republic

**Keywords:** neurosteroids, neuroactive steroids, cholesterol, muscarinic receptors, allosteric modulation

## Abstract

Muscarinic acetylcholine receptors are membrane receptors involved in many physiological processes. Malfunction of muscarinic signaling is a cause of various internal diseases, as well as psychiatric and neurologic conditions. Cholesterol, neurosteroids, neuroactive steroids, and steroid hormones are molecules of steroid origin that, besides having well-known genomic effects, also modulate membrane proteins including muscarinic acetylcholine receptors. Here, we review current knowledge on the allosteric modulation of muscarinic receptors by these steroids. We give a perspective on the research on the non-genomic effects of steroidal compounds on muscarinic receptors and drug development, with an aim to ultimately exploit such knowledge.

## 1. Introduction

Muscarinic acetylcholine receptors (mAChRs) are members of the G-protein-coupled receptor (GPCR) family, and are represented by five distinct receptor subtypes, M_1_–M_5_ [1]. When activated by their endogenous agonist acetylcholine (ACh), mAChRs exert their functions through second messenger cascades by coupling to specific classes of G-proteins (Figure 1). The M_1_, M_3_ and M_5_ subtypes preferentially activate phospholipase C (PLC) and promote calcium mobilization through G_q/11_, while M_2_ and M_4_ receptors inhibit the activity of adenylyl cyclase and thus cAMP synthesis via the G_i/o_ family of G-proteins [2].

Given the distribution of individual mAChRs subtypes, their expression levels and activation of distinct signaling cascades, these receptors play an important role in mediating a wide range of physiological functions in the central and peripheral nervous systems [3]. Malfunction or dysregulation of cholinergic signaling mediated by these receptors is strongly associated with the development of multiple pathological conditions and, as a consequence, targeting individual mAChRs subtypes represents a promising therapeutic approach for the treatment of neurologic and psychiatric conditions, e.g., Alzheimer’s disease, Parkinson’s disease, schizophrenia, substance abuse (for review see [4]) and diseases such as type 2 diabetes, asthma, cardiovascular diseases, and incontinence [5,6,7].

The mAChRs are also characterized by the highly conserved structure of their orthosteric binding sites; thus, it is virtually impossible to selectively activate individual subtypes of the receptor. This issue directed the research toward the development of compounds which act as allosteric modulators of mAChRs. By definition, allosteric modulators bind to a site spatially distinct from that of the endogenous transmitter; they change receptor conformation, leading to alterations in binding properties of the specific ligand (ACh for mAChRs), i.e., its affinity and ultimately potency and efficacy [9,10]. Allosteric binding sites of mAChR are far less conserved in their structures and thus offer a possibility for the development of mAChRs subtype-specific compounds [11]. Consequently, in the past decades, a great effort has been dedicated to the research and development of allosteric modulators that bind to these less conserved sites.

The most studied allosteric modulators of muscarinic receptors are neuromuscular blockers, such as gallamine [12,13], alcuronium and pancuronium [14]. According to the literature, many other allosteric modulators have been described. For example, thiochrome [15], verapamil [16], strychnine, [17], (−)-eburnamonine [18], fangicholine and tetrandrine [19], and 9-methoxy-α-lapachone [20], along with many others. The allosteric modulation of muscarinic receptors has been studied in a variety of pathological states [21,22,23]. As result of an enormous study, selective allosteric modulators have been identified. For example, benzylquinoline carboxylic acid (M1-selective allosteric modulators) [24,25] and compounds VU0010010, VU0152099, VU0152100, and LY2033298 (M4-selective allosteric modulators) have all been determined as such [26,27,28]. The selective positive allosteric modulation is considered a druggable target for the potential treatment of psychiatric and neurologic disorders, like Alzheimer’s disease or schizophrenia.

Multiple reports have shown that the interaction of steroids with mAChRs affects the ligand binding and functional responses of these receptors. It was demonstrated that molecules of membrane cholesterol (CHOL) change the affinity of muscarinic ligands and affect mAChRs activation dynamics [29,30,31]. CHOL represents a structural building block for all endogenous steroids, including steroid hormones (SHs) [32,33,34,35]. Typically, SHs (corticosteroids, sex steroids, bile acids) exert their well-known regulatory effects via activation of nuclear receptors (NRs), resulting in gene transcription and protein synthesis. This accounts for their long-lasting genomic effects from hours to days. It is noteworthy that SHs are also known to directly bind to and activate their specific membrane receptors, such as the progesterone receptor (mPR) or androgen receptor (mAR), along with estrogen membrane receptors α and β (mERα and mERβ), as well as the GPCR receptor GPR30 [36,37]. However, direct binding and modulation of actions of membrane-located receptors is a much faster process than the classic genomic effects, and lasts from milliseconds to minutes [36].

The non-genomic properties are characteristic of steroids synthesized de novo from the CHOL in the nervous system, in particular for neurosteroids (NSs) and also SHs [38]. Their synthetic analogues, which employ the same mode of action, are called neuroactive steroids (NASs). By acting through numerous ligand-gated ion channels, voltage-gated ion channels, or GPCRs, NSs produce various effects on the central and peripheral nervous system (CNS and PNS) [33,39]. Among NSs, SHs are also known to affect ACh release and cholinergic neurotransmission via interaction with mAChRs, improving memory and cognition [40,41]. Multiple reports describe that SHs bind to mAChRs and inhibit the interaction with muscarinic ligands at micromolar or higher concentrations [42,43,44,45], suggesting that steroids present in the body at physiological, i.e., nanomolar concentrations, cannot modulate mAChRs. However, it was recently proven otherwise. SHs can also act as NSs, i.e., progesterone and corticosterone, which have been shown to bind to mAChRs and modulate them in an allosteric manner at nanomolar concentrations [46]. Moreover, presumed binding sites of NSs and NAS at mAChRs were also identified [47].

There is a growing need for summarization and discussion of previous reports concerning the effects of steroids on mAChRs. Until now, an exact mechanism of action and a manner of steroids binding to these receptors has not been fully understood. In this review, therefore, we will focus on the mAChRs in the CNS and the allosteric modulation of their activity by endogenous and exogenous steroids. In particular, we describe the direct and indirect modulation of mAChRs by CHOL, SHs, NSs and NASs.

## 2. Cholesterol, Neurosteroids and Neuroactive Steroids

### 2.1. Cholesterol

A steroidal molecule is characterized by a tetracyclic cyclopenta[a]phenanthrene skeleton that has a specific position numbering and ring letters (Figure 2). The primary function of CHOL is structural. It serves as the main building block for synthesizing various SHs (gonadal sex hormones and adrenal glucocorticoids and mineralocorticoids), vitamin D, bile acids, and also NSs. A simplified scheme of steroid biosynthesis is summarized in Figure 3.

CHOL is also an essential component of the cell membranes, maintaining their fluidity and integrity. Within the membrane, a polar C-3 hydroxyl group of CHOL interacts with surrounding phospholipids and proteins, while the tetracyclic steroid skeleton with the lipophilic C-17 substituent interacts with the fatty acids. This enables the integration of CHOL into the lipid bilayer and secures its integrity [48]. Molecules of CHOL are distributed throughout the plasma membranes and may form dimers, as detected in X-ray crystal structures of membrane proteins [49], or concentrate in specialized sphingolipid-rich domains known as lipid rafts [50]. Fractions of membranes rich in CHOL are thicker and more rigid. Near the receptors, lipid rafts interfere with the machinery of signal transduction [50], diminishing the availability of signaling molecules and affecting the activity of the receptors [51]. Thus, the close interaction of CHOL with membrane proteins, including ion channels and GPCRs, affects processes of ligand binding, receptor activation and signal transduction [52,53,54].

Lipid molecules are frequently found in X-ray and cryo-EM structures of GPCRs, indicating that these lipids may specifically interact with GPCRs in their membrane environment [54]. If so, then they may allosterically modulate ligand binding to the receptors and the functional response of receptors to agonists. As CHOL binds to multiple specific binding sites on many GPCRs, it can be considered their allosteric modulator [55,56,57]. Indeed, membrane CHOL was found to co-crystallize with various GPCRs for distinct classes of agonists as published in the RCSB database (https://www.rcsb.org/ (accessed on 20 February 2022)). As for the mAChRs, CHOL was not found in the crystal structures of the receptors. However, its binding site was revealed using molecular docking [31]. The molecules of CHOL can directly influence GPCR activity by altering the binding of a specific ligand, affecting receptor activation as well as a receptor-to-G-protein coupling. CHOL can also affect GPCRs indirectly through changes in the membrane organization, such as alterations in the fluidity of the membrane surrounding the receptor and thus effectors available for signaling (signal trafficking). For review, see [10,58,59].

### 2.2. NSs, NASs–Functions and Their Genomic and Non-Genomic Effects

As mentioned previously, NSs represent a class of endogenous compounds synthesized de novo in CNS from CHOL or steroidal precursors imported from peripheral endocrine glands. NSs are known to modulate neuronal excitability by acting through various ligand-gated ion channels and GPCRs [38,54], both, positively and/or negatively. The best-known function of NSs in CNS is the modulation of γ-aminobutyric acid (GABA_A_) receptors responsible for inhibitory neurotransmission in the brain [60]. Further, some NSs modulate the *N*-methyl-*D*-aspartate (NMDA) glutamate receptors, α-amino-3-hydroxy-5-methyl-4-isoxazolepropionate (AMPA)/kainate receptors, glycine or nicotinic acetylcholine receptors (nAChRs) [33]. Consequently, NSs are involved in the regulation of various CNS functions such as cognition and memory processes [61]. Moreover, NSs also modulate pain pathways [62,63,64,65], and exert neuroprotective effects [66,67,68], among a myriad of other actions. For review, see [69].

In contrast, SHs, by definition, should exert multiple functions via the activation of nuclear receptors (NRs) specific for steroid hormones [70]. Typically, activation of NRs induces gene transcription and protein synthesis, therefore their action is relatively slow (hours to days) [36]. Nevertheless, there is a piece of strong evidence that, in addition to the classical genomic mechanism of action, SHs can exert rapid, non-genomic signaling via interaction with membrane receptors [36,71,72]. Consequently, the literature describes the crucial role of SHs in the development and functioning of the CNS [72,73,74,75]. For example, synthesized locally within CNS, progesterone [76] and estradiol [77] influence neuronal functions and produce a variety of effects that are unrelated to reproduction [78,79,80,81]. Acting as NSs, estradiol, progesterone as well as corticosterone, regulate cognition [82,83,84], memory [85,86], brain development [87] and behaviour [88].

On the other hand, some steroids share both hormonal and neurosteroid activity. For example, allopregnanolone—a well-known example of a potent allosteric modulator of GABA_A_ receptors—was demonstrated to also exert genomic effects via activation of mPRs [89,90]. Similarly, dehydroepiandrosterone (DHEA), a metabolic intermediate in the biosynthesis of many SHs and the most abundant hormone in mammals, is secreted by the adrenal gland and ovary. Its hormonal effects are mediated through androgen and estrogen receptors, peroxisome proliferator-activated receptor (PPAR), pregnane X receptor (PXR), and the constitutive androstane receptor (CAR) [91]. Regarding neurosteroid activity of DHEA, it has been described as a ligand of GABA_A_, NMDA, sigma-1 receptors and L-type calcium channels. This explains the effects of DHEA on physiological functions and pathological abnormalities in the brain [92,93].

Taken together, the current literature shows that the effects of steroids are complex and cannot be assigned to a single mode of action. Such a multi-target mode of action may explain their unique drug-likeness in a variety of neurological and psychiatric conditions. When interacting with membrane receptors, the effects of NSs are exerted in a manner of non-genomic signaling. However, chronic exposure to NSs may indirectly (non-genomic pathways) induce genomic action (e.g., changes in receptor expression) [33,94]. Therefore, the crosstalk between genomic and non-genomic steroid effects needs to be taken into consideration. The interplay of genomic and non-genomic actions of NASs is summarised in Figure 4.

## 3. Muscarinic Receptors

Muscarinic acetylcholine receptors (mAChR) are members of class A, Rhodopsine-like GPCRs, and are represented by five distinct receptor subtypes, M_1_–M_5_ [1]. Like all GPCRs, mAChRs are integral membrane proteins consisting of seven transmembrane α-helices (TM1 to TM7) connected via three intracellular (ICL1 to ICL3) and three extracellular (ECL1 to ECL3) loops (Figure 1). Individual TM helixes form a hydrophilic pocket (orthosteric binding site) accessible from the extracellular side for endogenous signaling molecules. mAChRs activation by their endogenous agonist ACh results in subsequent G-protein activation, and depending on the G-protein class, mediates various cellular responses [97,98].

Activated mAChRs trigger distinct second messenger cascades coupled to designated G-protein classes and thus mediate a wide range of physiological functions throughout the body. M_1_, M_3_, and M_5_ mAChR subtypes preferentially activate G_q/11_ G-proteins to stimulate phospholipase C (PLC) and induce the mobilisation of intracellular calcium. M_2_ and M_4_ receptors activate G_i/o_ G-proteins, the α-subunit of which inhibits adenylyl cyclase (AC), decreasing the production of cAMP, while the βγ-dimer of G-proteins modulates conductance of K^+^ and Ca^2+^ channels [3,99]. Besides these preferential signaling pathways, muscarinic agonists may activate also other ones, termed non-preferential signaling pathways [100,101].

Specific targeting of mAChRs subtypes, and thus selective regulation of their signaling pathways, might be of great value in seeking the treatment for the CNS [4] and diseases affecting internal organs [5,6,7].

## 4. Direct Effects of Steroids on mAChRs

### 4.1. Direct Effects of Cholesterol

Molecules of CHOL can directly influence GPCRs by altering the binding of a specific ligand, activation of a receptor as well as preferences of receptor-to-G-protein coupling. In contrast, CHOL can affect GPCRs indirectly through changes in the membrane organization, such as alterations in the fluidity of the membranes surrounding GPCRs and thus signal trafficking [58,59].

The membrane CHOL modulates GPCRs by acting on their allosteric binding sites. CHOL-binding motifs were predicted based on analyses of X-ray and cryo-EM structures of various GPCRs. Three CHOL-binding motifs were described so far. The motif common to all membrane proteins is the Cholesterol Recognition Amino acid Consensus (CRAC) [102] and its inverse variant (CARC) [103]. The so-called Cholesterol Consensus Motif (CCM), the groove formed by the transmembrane domains TM2, TM3 and TM4, was identified in the structure of the β_2_-adrenergic receptor [56]. As CHOL-binding sites on GPCRs are distinct from the binding sites of endogenous transmitters, CHOL can be considered an allosteric modulator [56]. Allosteric binding sites on mAChRs represent far less conserved structures compared to their orthosteric binding site, and offer a possibility to target specific receptor subtypes [11].

Initially, it was demonstrated that CHOL directly affects the affinity of muscarinic ligands. An increase in the content of CHOL within the membrane resulted in a reduced affinity for the muscarinic agonist carbachol at M_2_, but increased its affinity at M_1_ and M_3_ receptors. On the other hand, CHOL depletion increased the affinity of carbachol to M_1_, M_2_, and M_3_ subtypes. In contrast, CHOL depletion was shown to diminish the affinity of the muscarinic antagonist N-methylscopolamine (NMS) at these receptors. Enrichment of membranes with CHOL caused a decrease in affinity for NMS at M_1_ and M_3_, and an increase in affinity at the M_2_ receptor [29,30].

Changes in the content of the membrane CHOL also affect preferential and non-preferential signaling through the M_2_ as well as M_1_, and M_3_ expressed in CHO cells [29,30]. CHOL-dependent changes in preferential mAChRs signaling are presented in Figure 5. Regarding M_2_, CHOL depletion significantly strengthens the preferential signaling pathway G_i/o_ and reinforces the maximal effect of inhibition of cAMP synthesis. It also stimulated non-preferential G_s_ and G_q/11_ signaling pathways, as shown in Figure 5B. Additionally, in the case of M_1_ and M_3_ receptors, both gradually increase and decrease in membrane CHOL concentration, resulting in a concentration-dependent inhibition of the preferential signaling pathway via G_q/11_ and a decrease in accumulation of inositol trisphosphate(Figure 5B,C). As for the non-preferential G_s_-mediated signaling, an increase in membrane CHOL concentration inhibited the cAMP accumulation, while a decrease in membrane CHOL concentration stimulated cAMP synthesis.

Based on molecular modeling, the CHOL allosteric binding site was found in the intracellular leaflet of the membrane between TM6 and TM7 of mAChRs [31]. This binding site presumably also represents a site of binding for various steroidal compounds. In addition, subtype specificity of some ligands was shown to be affected by the content of membrane CHOL. Specifically, binding of CHOL at the TM6 and TM7 interface attenuates activation of M_1_, M_4_ and M_5_ receptors [31].

Xanomeline (3-(hexyloxy)-4-(1-methyl-1,2,5,6-tetrahydropyridin-3-yl)-1,2,5-thiadiazole) is a muscarinic agonist considered functionally selective for M_1_ and M_4_ receptors, developed for treatment of Alzheimer’s disease [104,105]. Xanomeline binding to mAChRs is partially resistant to washing [106]. Wash-resistant xanomeline steadily activates mAChRs with an exception of the M_5_ subtype [107]. Mutation of leucine 6.46 to isoleucine at the CHOL binding site in M_1_ and M_4_ receptors resulted in receptors insensitive to activation by wash-resistant xanomeline. On the other hand, the mutation of isoleucine 6.46 to leucine in the M_5_ receptor made it sensitive to activation by wash-resistant xanomeline. Decreasing membrane CHOL content reversed the effects of mutations, indicating that xanomeline functional selectivity is rather the result of specific receptor–membrane interactions than agonist–receptor interactions [31]. Thus, changing membrane CHOL level or interaction of a receptor with the membrane might represent a novel possibility to achieve pharmacological selectivity for mAChRs.

### 4.2. Non-Genomic Effects of NSs and NASs on mAChRs

Historically, multiple reports showed that the interaction of SHs with mAChRs alters the binding dynamics of various ligands [42,43,44,108,109,110,111]. Early research conducted on the direct effects of SHs on mAChRs was focused on changes in the binding of radiolabelled muscarinic antagonists like [^3^H]-quinuclidinyl benzilate ([^3^H](-)QNB), N-methyl-[^3^H]-4-piperidyl benzilate ([^3^H]4NMPB) or N-methyl-[^3^H]-scopolamine ([^3^H]NMS) in the cell membranes prepared from rat brain tissues. In competitive binding studies with [^3^H]4NMPB, the steroids progesterone and estradiol affected the binding properties of the mAChR agonist, oxotremorine. Both steroids decreased the affinity and proportion of the high-affinity binding sites [112]. Later in experiments with [^3^H]NMS, other researchers confirmed that progesterone and estradiol (but not testosterone) bind to mAChRs in the membranes prepared from the rat hypothalamus and amygdala tissues [113].

In the study by Klangkagaya and Chan [44], the effects of 50 steroid compounds on [^3^H](-)QNB binding to mAChRs in hypothalamic and pituitary membranes was reported. The structures of active pregnane and androstane compounds, including their IC_50_ values in inhibiting the [^3^H](-)QNB binding, are summarized in Figure 6 and Figure 7, respectively. The results of this study demonstrated that the pregnane skeleton is considerably more relevant for further development than the androstane skeleton. Further, it was determined that incomplete inhibition of [^3^H](-)QNB binding by tested steroidal compounds indicates allosteric modulation of [^3^H](-)QNB binding [44].

Regarding the structure–activity relationship, modifications of the progesterone skeleton afforded structures with IC_50_ values in the tens of µM. In contrast, except for testosterone acetate with an IC_50_ value of 18 µM, the IC_50_ values of androstane analogues varied from 100 to 200 µM. Interestingly, the 17α-hydroxy-substituted pregnane skeletons were active, except for hydrocortisone and 17α-hydroxy-5α-pregnan-3,20-dione (IC_50_ > 200 µM). Similarly, the hydroxylation of the skeleton in position C-21 was tolerated well. In contrast, hydroxylation at position C-11 strongly diminished the affinity for mAChRs (corticosterone and hydrocortisone, IC_50_ > 200 µM), while the presence of the 11-oxo group decreased the affinity only slightly. Reduction of the Δ^3,4^-enone to 5β-steroids afforded compounds with higher affinity than the corresponding 5α-analogues. Accordingly, the orientation of the hydrogen atom at the C-5 position was identified as crucial for the inhibition of [^3^H](-)QNB binding.

As mentioned previously, modification of the testosterone skeleton did not afford active compounds with low micromolar affinities (Figure 7). Out of 50 tested compounds, 16 of them were androstanes, and 7 of them showed the ability to inhibit [^3^H](-)QNB binding to mAChRs. It should be mentioned that estradiol was also inactive (IC_50_ > 200 µM) [44].

Further, the allosteric mode of mAChRs modulation by NSs and NASs was described in the study of Shiraishi et al. [111]. The synthetic analgesic neurosteroid alfaxalone (3α-hydroxy-5β-pregnane-11,20-dione) decreased [^3^H](-)QNB binding to M_1_ and M_3_ receptors (IC_50_ 2.6 μM and 4.5 μM, respectively) and inhibited acetylcholine-induced Ca^2+^-activated Cl^−^ currents in oocytes, expressing M_1_ and M_3_ receptors (IC_50_ values of 1.8 µM and 5.3 µM, respectively). A selective protein kinase C inhibitor GF109203X had a negligible effect on the inhibition of ACh-induced currents by alfaxalone, confirming allosteric modulation of these mAChRs [111].

Similarly, Horishita et al. [42] described voltage clamp experiments showing that pregnenolone and progesterone altered acetylcholine-induced Ca^2+^-activated Cl^−^ currents, while DHEA did not affect the function of M_1_ and M_3_ receptors expressed in Xenopus oocytes. The IC_50_ values at M_1_ and M_3_ for progesterone were 2.5 and 3 μM, while for pregnenolone they were 11.4 and 6 μM, respectively. Further, both steroids were shown to diminish [^3^H](-)QNB binding to M_1_ and M_3_ receptors. Both steroids also affected affinity and binding capacity, indicating non-competitive inhibition, showing that tested steroids bind to M_1_ and M_3_ receptors at an allosteric binding site [42].

The literature summarized above describes the effects of steroids on specific ligand binding to muscarinic mAChRs. In general, these binding studies suggested that compounds sharing a steroid scaffold with CHOL bind to mAChRs at the site distinct from their natural agonist ACh or radiolabelled orthosteric ligands used in these experiments. Yet, except for the studies on the alteration of ACh-mediated responses in Xenopus oocytes [42,111], evidence gained from these reports accounted mainly for the manner of ligand–receptor interaction. Moreover, these reports describe the effects of steroids on mAChRs at micromolar or higher concentrations, which exclude their physiological relevance but pointed to their pharmacological potential.

The authors of the recent study examined the allosteric modulation of mAChRs by 20 steroidal compounds (Figure 8) [46]. This study revealed that all tested compounds changed [^3^H]NMS equilibrium binding at a 10 μM concentration. Moreover, some compounds exerted high-affinity binding with sub-micromolar affinity. Importantly, corticosterone and progesterone were found to bind to the mAChRs with about 100 nM affinity, which is within the physiological range [114,115]. In particular, the structure–activity relationship evaluation of the results [46] has shown that some of the compounds with the highest affinities to mAChRs have an enone group (3-oxo-4-ene structure) in the A-ring. Further, the 5β-steroids generally have higher affinities to all receptor subtypes than their 5α-analogues. The presence of the C-17 acetyl group was shown to represent a key structural element for affinity improvement. Corticosterone with hydroxyl groups at C-11 and C-21 had a higher binding affinity, while the presence and orientation of hydroxyl moiety at C-3 had no significant effect. The aromatization of A-ring, such as the formation of estradiol from testosterone, ended the ability of a compound to affect [^3^H]NMS binding to mAChR. These findings are in agreement with the structural features of steroids that diminish [^3^H](-)QNB binding to hypothalamic mAChR [44].

Four compounds, in particular corticosterone, progesterone, estradiol and testosterone, affected the functional response of mAChRs at physiologically relevant concentrations. The influence of steroids on mAChRs functional response to ACh was quantified by changes in specific [^35^S] GTPγS binding to membranes expressing M_2_ or M_4_, or inositol phosphates accumulation in CHO cells expressing M_1_, M_3_ or M_5_ receptors. Corticosterone induced a 3-fold increase in ACh potency at M_2_, but decreased it 3-fold at the M_4_ receptor. Progesterone increased the efficacy of ACh receptors by 30% at M_1_ and by 20% at M_3_ receptors and decreased it by 30% at M_2_ receptors. Estradiol increased the efficacy of ACh by 24% at the M_1_ receptor.

The follow-up study [47] describes the binding site on muscarinic acetylcholine receptors for NASs (Figure 9). It was found that NASs can bind to the two distinct allosteric binding sites on mAChRs, with approximately 100 nM and 10 μM affinities. The high-affinity binding site was investigated in [^3^H]NMS binding experiments using selected NAS in combination with well-known classic muscarinic allosteric modulators gallamine and alcuronium, and steroid allosteric modulators pancuronium, rapacuronium and WIN-compounds [47]. This high-affinity binding site was shown to be different from the common, extracellularly located allosteric binding site for alcuronium or gallamine, or the aminosteroid-based muscle relaxants pancuronium and rapacuronium. Interestingly, selected NAS bound to the same high-affinity binding site as steroid-based WIN-compounds that do not bind to the classical allosteric binding site located between the ECL2 and ECL3 [116,117,118,119,120]. Compounds 5α-androst-1-en-17β-yl 17-hemisuccinate (MS-96) and 17-methylene-5β-androstan-3α-yl 3-hemiglutarate (MS-112) were able to bind to this site with an affinity of about 50 nM and 16 nM, respectively. The authors have also shown that the membrane CHOL competes with NASs and WIN-compounds for binding to both high- and low-affinity binding sites. It suggests that the high-affinity binding site is rather oriented towards the inner side of the membrane, and that this site may represent a novel target for the allosteric modulation of muscarinic receptors. However, identification of the exact number and location of the CHOL binding sites at mAChRs remains to be determined [47,121].

## 5. Conclusions

Steroidal compounds such as cholesterol, neurosteroids, neuroactive steroids and steroid hormones bind to several sites on muscarinic acetylcholine receptors. From these sites, they allosterically modulate the binding of muscarinic ligands and the functional response of muscarinic receptors. They share a common high-affinity binding site that is oriented towards the membrane. Neurosteroids and steroid hormones allosterically modulate muscarinic receptors at physiologically relevant concentrations. The physiological non-genomic effects of neurosteroids and steroid hormones have not been studied in detail so far.

## 6. Perspectives

Allosteric modulation of muscarinic receptors by steroids proposes two new avenues for future research. One is an exploration of the physiology of the non-genomic effects of neurosteroids and steroid hormones at muscarinic receptors. Besides novel knowledge, an understanding of the non-genomic effects of neurosteroids and steroid hormones may bring new ways for the treatment of diseases resulting from a malfunction of muscarinic signaling by manipulation with levels of neurosteroids or steroid hormones. The other is exploiting differences in receptor–membrane interactions for the development of selective modulators. These differences can be approached in two ways. First, differences in receptor–membrane interactions among receptor subtypes allow the development of subtype-selective compounds. Second, differences in receptor–membrane interactions among various tissues give the opportunity for tissue-specific modulation. For example, drugs targeting cholesterol binding sites will be more efficient at cholesterol-lean membranes than at cholesterol-rich ones due to competition with membrane cholesterol.

## Figures and Tables

**Figure 1 ijms-23-13075-f001:**
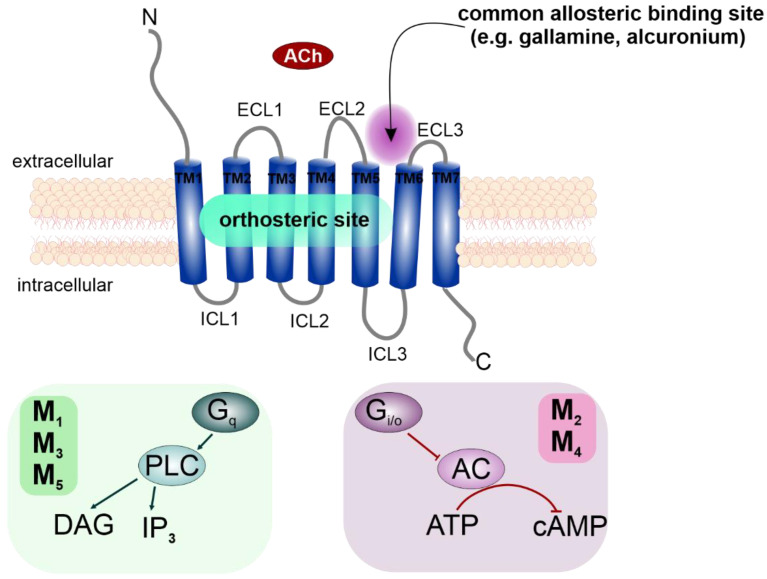
**Schematic structure of mAChRs and their preferential signaling pathways.** Classical allosteric modulators of mAChRs, such as gallamine or alcuronium, bind to the extracellular part of the receptor between ECL2 and ECL3 [8]. Ach—acetylcholine; ECL—extracellular loop; TM—transmembrane α-helix; ICL—intracellular loop; DAG—diacylglycerol; IP_3_—inositol triphosphate; AC—adenylyl cyclase; PLC—phospholipase A.

**Figure 2 ijms-23-13075-f002:**
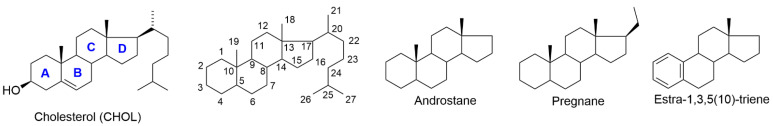
Structure of cholesterol with ring letters (A–D), ring numbering (1–27) and trivial names of basic skeletons relevant for this review.

**Figure 3 ijms-23-13075-f003:**
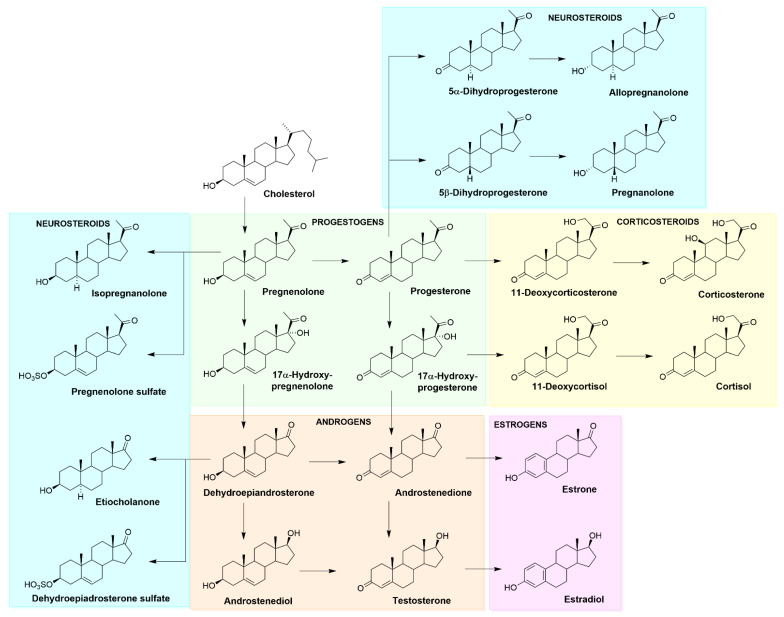
Simplified scheme of steroid biosynthesis, including the major classes of steroid hormones and examples of neurosteroids.

**Figure 4 ijms-23-13075-f004:**
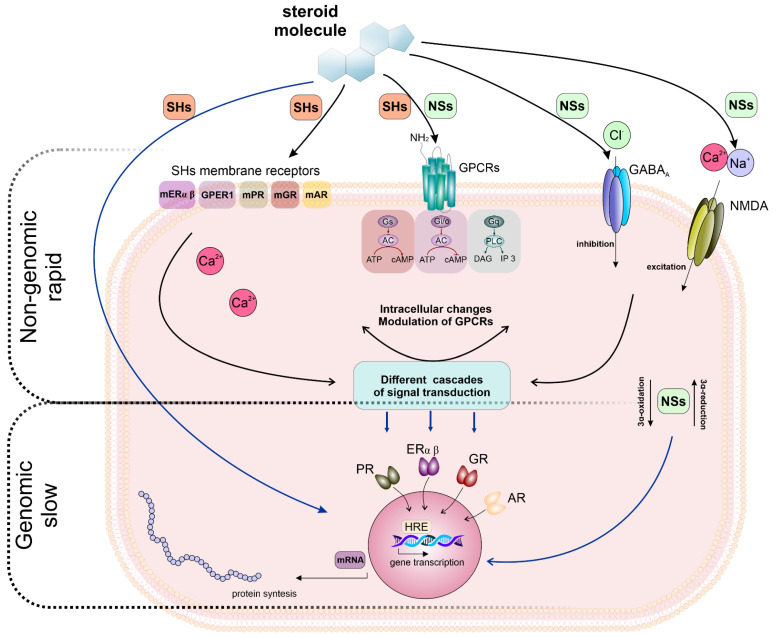
The interplay between genomic and non-genomic effects of steroids. **Top**, non-genomic, rapid (seconds to minutes) signaling: the activation of membrane-localized receptors for estrogen (mERα; mERβ; GPER1/GPR30), progesterone (mPR), glucocorticoids (mGR) and androgens (mAR) by a specific hormone modulates numerous signaling cascades and produces different cellular effects [95,96]. NSs exert their rapid, non-genomic effects via modulation of membrane ionotropic receptors and channels, e.g., γ-aminobutyric acid receptors, GABA_A_ or NMDA receptors, resulting in excitability changes within neurons. Activation of G_s_ protein results in stimulation of adenylyl cyclase (AC) and cAMP synthesis; activation of G_q/11_ protein results in the activation of phospholipase C (PLC) and production of inositol 1,4,5-trisphosphate (IP_3_) and diacylglycerol (DAG); activation of G_i/o_ proteins inhibits AC and cAMP synthesis. **Middle**, metabolites of neurosteroid (NSs) produced by intracellular oxidation bind to steroid receptors [33,35]. **Bottom,** slow genomic effects (minutes to hours): steroid hormones (SHs) bind to their specific intracellular class I nuclear receptors (progesterone receptor (PR), oestrogen receptors (ERα and β), glucocorticoid receptors (GR) or androgen receptors (AR)) which, in the absence of the ligand, reside in the cytosol. The binding of the ligand to these receptors results in the translocation of the receptor–ligand complex to the nucleus where it binds to specific hormone response elements (HREs) and regulates gene transcription. The figure was prepared according to [33,36,94].

**Figure 5 ijms-23-13075-f005:**
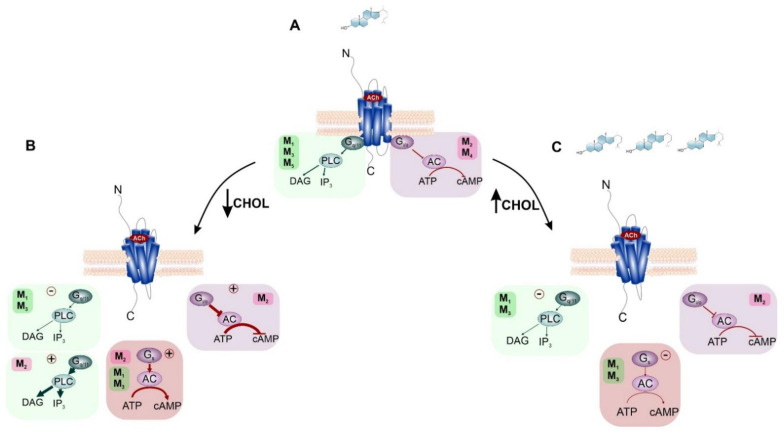
Influence of membrane CHOL content on preferential and non-preferential signaling of mAChRs. (**A**) standard mAChRs signaling in the cell membrane with a natural level of CHOL (light blue steroid molecule). Preferential coupling of M_1_, M_3_, M_5_ receptors to G_q/11_ protein (green box) and M_2_, M_4_ receptors coupling to G_i/o_ protein (purple box). (**B**) depletion of membrane CHOL diminishes preferential G_q/11_ signaling and enhances preferential G_i/o_ and non-preferential G_s_ (red box) and G_q/11_ signaling. (**C**) increase in membrane CHOL level attenuates signaling via preferential G-proteins.

**Figure 6 ijms-23-13075-f006:**
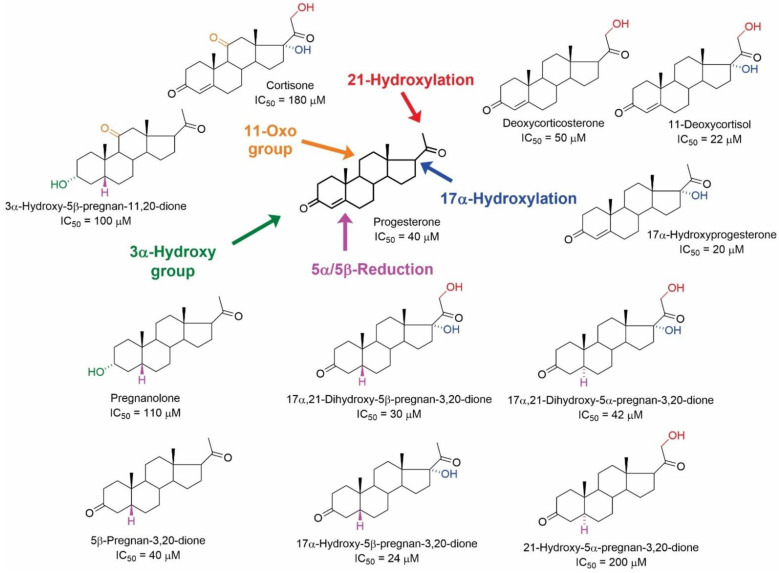
Structures of pregnane steroids with their IC_50_ values in inhibiting the [^3^H](-)QNB binding from the study of Klangkalya and Chan [44].

**Figure 7 ijms-23-13075-f007:**
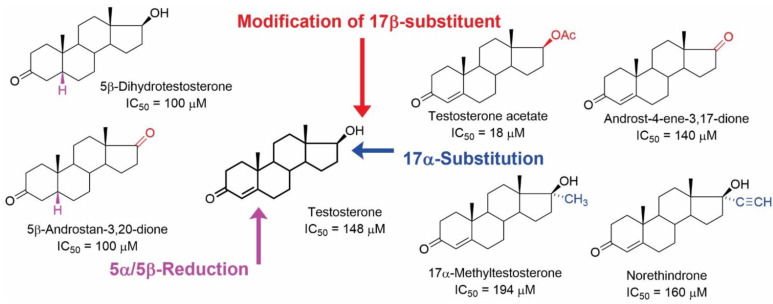
Structures of androstane steroids with their IC_50_ values in inhibiting the [^3^H](-)QNB binding from the study of Klangkalya and Chan [44].

**Figure 8 ijms-23-13075-f008:**
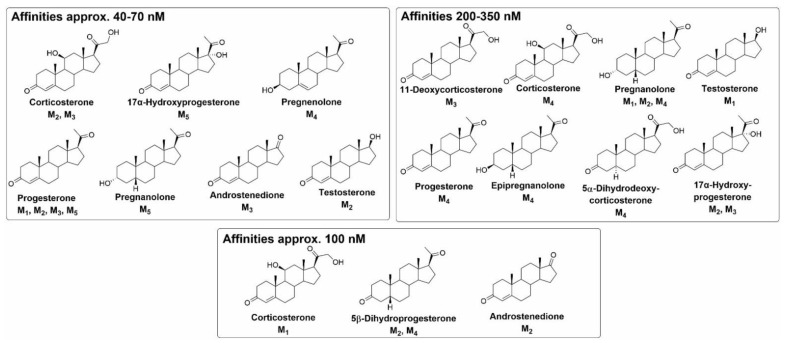
Overview of binding affinities to individual muscarinic receptor subtypes from [46].

**Figure 9 ijms-23-13075-f009:**
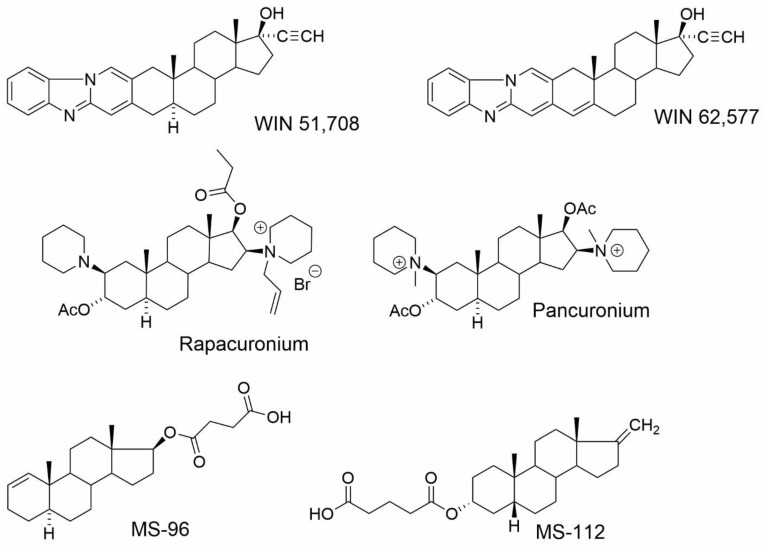
Structures of WIN compounds, the skeletal muscle relaxants pancuronium and rapacuronium and NASs tested in the study of Dolejsi et al. [47].

## Data Availability

Not applicable.

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
