# Peer review of "Allosteric Modulation of Muscarinic Receptors by Cholesterol, Neurosteroids and Neuroactive Steroids"

_ijms, 2022, doi:10.3390/ijms232113075_

Round 1

Reviewer 1 Report

The manuscript "Allosteric modulation of muscarinic receptors by cholesterol, 2 neurosteroids and neuroactive steroids" by Ewa Szczurowska et al. is a short review of known facts about possible modulation of activity of muscarinic receptors strongly related to human physiology and health. Muionscarinic G Protein Coupled Receptors are critical in regulation of so many cellular procesess that they are objects of intensive studies. Numarous drugs act on GPCRs (M type as well) so the topic of this manuscript is very well justified and may be interested to many readers of IJMS.

Overall presentation is clear and logical, though sometimes only very basic facts are given regarding paricular compounds/receptor interactions. The authors concentrate on allosteric modulation (more selective/specific sites than orthosteric ones) by Cholesterol, neurosteroids and neuroactive steroids. Such limited set of modulators is justified, though more general discussion (a short paragraph) on other possible chemicals interactions with allosteric muscarinic receptors might increase the value of this manuscript further. Of course, the literature on this topic is not covered in full but this is a right of the authors.

I have only several minor remarks:

1. Abstract: "Cholesterol, neurosteroids, neuroactive steroids, and steroid 11 hormones are molecules of steroid origin that, besides having notoriously known genomic effects"
Is "notoriously" the best and intended word here?

2. In fig 1. There is a typo "Exctracellular". In Fig.1 Caption: AC not explained , PLC worth repeting from the texts in figure legend  

3. Page 2, line 52 . Is CHOL a simple molecule? "was demonstrated that even a simple molecule of membrane cholesterol (CHOL) changes the affinity" (? single?)

4. Page 3, line 78: "these receptors is not been fully understood." check grammar usage (?have?)

5. Page 4, lines 109-110: "Lipid molecules are frequently found in X-ray and cryo-EM structures of GPCRs, indicating that these lipids regulate GPCRs in their membrane environment via specific in-110 teractions"
In my opionion is is overstatement or simplification: from the mere presence of lipids in Xray structures regulations of GPCRs by lipids cannot be directly infered. Re-phrase.

6.Fig.5, (C) "increase in membrane CLR level attenuates signalling via preferential G-proteins" CLR or CHOL?

. Page 8 line 251" "The allosteric binding site for CHOL was found in the intracellular leaflet of the mem-251 brane between TM6 and TM7 of mAChRs" Check intra- or extra- ???

Reviewer 2 Report

By the Title, the review by Szczurowska and coworkers promises to discuss the allosteric modulation of muscarinic receptors by cholesterol derived compounds such as neurosteroids and neuroactive steroids.

The authors describe how cholesterol and cholesterol derived molecules can interact with mACHR at allosteric sites and modulate their molecular signaling pathway cascades in a receptor subtype manner. They further review and highlight the importance of gathering new knowledge on the non-genomic effects of steroidal compounds on mAChR.

I have some comments to the authors that may contribute to the final version of the manuscript.

In the abstract the authors use the word notoriously (line 12) to emphasize the known genomic effects of steroids. However, I suggest to change this word because its meaning is related to a negative or bad quality or fact.

The authors first describe the biophysics and physiology of mAChR. However, through lines 30 to 36 the authors mention almost using the same words as in lines 200-204 the possible therapeutic approaches for pathological states where mAChR are implicated. I am sure the authors can rewrite these lines so that they are not identical.

Lines 35 and 203: what are internal diseases? I believe the authors should change internal for metabolic.

Line 23: add the word "endogenous" before agonist acetylcholine.

Line 25: a word is missing “… (promote) calcium mobilization through…

Line 40: IP3, inositol trisphosphate.

Line 62: add reference.

Lines 79-81: Something is missing. The sentence is not complete. Including what?

Figure 2:

Please remove the "H". It confuses the reader since it looks like the authors are describing another ring…

Then in figure 3 this "H" is not shown so it can confuse the reader

Line 125: G protein-coupled receptors (GPCRs) was already abbreviated in lines 21 and 22 when it was first mentioned. Please remove.

Line 153: dehydroepiandrosterone (DHEA), please abbreviate before in line 148.

Figure 4: in light blue there is a scheme of a steroid. Please put the name next to the scheme.

In the legend of figure 4, line  177 it says: specific intracellular nuclear receptors (NRs). I could not find NRs on the figure. Instead I found PR, ER ab, AR, GR. Please correct and write what do these intracellular receptors represent.

Line 250: CLR is CHOL?

Line 256: add reference.

Many compound that interact with mAChR are mentioned in the manuscript. However, a description of their potential use in therapeutic is lacking. For example, in line 257 Xanomeline is mentioned in the text but it would enrich the reader to know that this drug was initially developed for the treatment of Alzheimer's disease (Bymaster et al., 1998) and has further been suggested for the treatment of schizophrenia (Shekhar et al., 2008), etc. Besides the biophysics interaction with the different subtypes of mAChR, the addition of these clinical data will further add light on why it is so important to achieve pharmacological selectivity for mAChRs for therapeutics.

High cholesterol levels have been proposed as a risk factor in the pathogenesis of neurological diseases such as AD and cardiovascular disease. Many people take statins to lower the levels of cholesterol. How do the authors believe these drugs may affect the allosteric modulation on the different subtypes of mAChRs. Please comment.

In the acknowledgment section the authors state that they created the Graphical abstract using BioRender.com. I did not see the graphical abstract. Weren’t figures 1, 4 and 5 also created using BioRender?
